# Atlas3D: Physically Constrained Self-Supporting Text-to-3D for Simulation and Fabrication

**Yunuo Chen**[1]*, **Tianyi Xie**[1]*, **Zeshun Zong**[1]*, **Xuan Li**[1],
**Feng Gao**[2]†, **Yin Yang**[3], **Ying Nian Wu**[1], **Chenfanfu Jiang**[1]
[1]University of California, Los Angeles, [2] Amazon, [3] University of Utah
{yunuoch, tianyixie77, zeshunzong, xuanli1}@ucla.edu, fenggo@amazon.com,
yin.yang@utah.edu, ywu@stat.ucla.edu, cffjiang@ucla.edu

## Abstract

Existing diffusion-based text-to-3D generation methods primarily focus on producing visually realistic shapes and appearances, often neglecting the physical constraints necessary for downstream tasks. Generated models frequently fail to maintain balance when placed in physics-based simulations or 3D-printed. This balance is crucial for satisfying user design intentions in interactive gaming, embodied AI, and robotics, where stable models are needed for reliable interaction. Additionally, stable models ensure that 3D-printed objects, such as figurines for home decoration, can stand on their own without requiring additional support. To fill this gap, we introduce Atlas3D, an automatic and easy-to-implement method that enhances existing Score Distillation Sampling (SDS)-based text-to-3D tools. Atlas3D ensures the generation of self-supporting 3D models that adhere to physical laws of stability under gravity, contact, and friction. Our approach combines a novel differentiable simulation-based loss function with physically inspired regularization, serving as either a refinement or a post-processing module for existing frameworks. We verify Atlas3D's efficacy through extensive generation tasks and validate the resulting 3D models in both simulated and real-world environments.

## 1 Introduction

Generating high-quality 3D content is of great importance in modern visual computing. Realistic 3D models are highly sought after in computer graphics, while robust real 3D assets are gaining attention in training embodied AI. Nevertheless, the standability of 3D models – the ability to stand steadily without additional support – is often neglected. Real-world man-made objects such as action figures, toys, and furniture inherently possess some degree of geometric stability, allowing them to be safely placed on the ground. Although one usually takes such standability for granted, existing generative models fail to produce steady 3D assets due to their lack of physical perception; see Fig. 1.

Incorporating this stability expectation into 3D generation will significantly reduce the human effort required for tasks such as sorting out unqualified meshes, post-processing geometries, or adding external supports before actually using the 3D asset in any simulator or the real world. Furthermore, creating physically plausible 3D content will enhance the fidelity of simulations and policy training with these objects, potentially narrowing the sim-to-real gap and empowering embodied AI in robotic tasks. Towards this goal, we develop a 3D generation framework that can produce high-quality models adhering to basic physical laws, such as gravity, stability, and frictional contact.

Several attempts have been made to incorporate physical constraints into 3D generation. Yang et al. utilized the spatial and physical sense of LLM to design floor plans and furniture arrangements [98].

---

*Equal contribution.
†This work is not related to F. Gao's position at Amazon.

38th Conference on Neural Information Processing Systems (NeurIPS 2024).

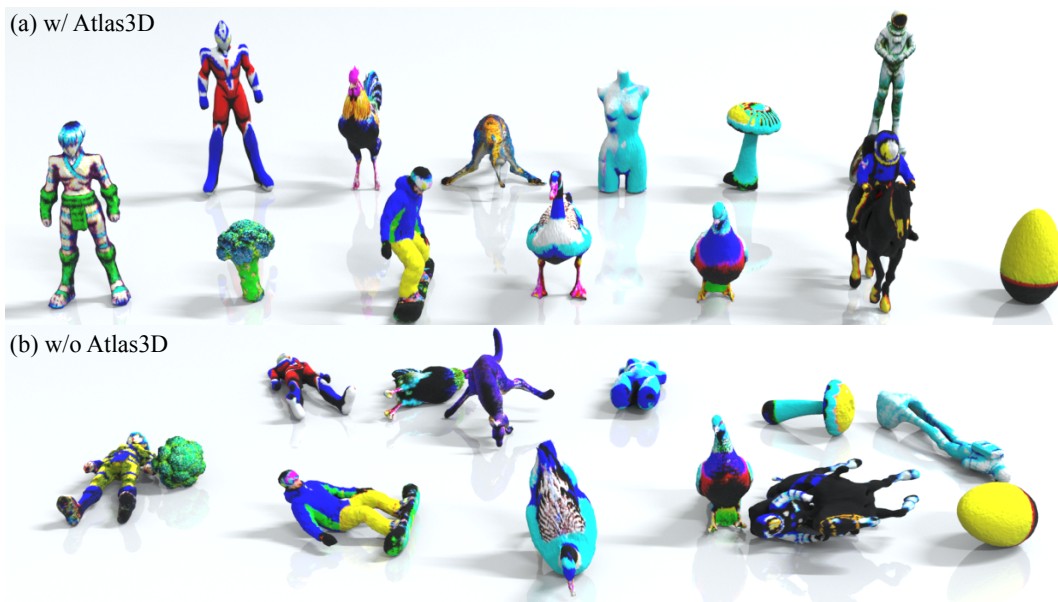

(a) w/ Atlas3D

(b) w/o Atlas3D

Figure 1: Simulation in ABD [27]: (a) 3D models generated from our Atlas3D framework can stand steadily on the ground; (b) those generated from existing methods tend to fall over.

PhyScene introduced physical guidance, such as collision and reachability constraints, to diffusion models to generate furniture layouts [97]. However, both works primarily consider straightforward spatial constraints, such as non-collision, and fail to incorporate more complex physics. Mezghanni et al. proposed a GAN-based network to generate physically-aware geometries by training a neural stability predictor using datasets labeled by Bullet [48]. Another GAN-based work by Wang et al. employed CFD software to compute vehicle drag coefficients, guiding the generation of streamlined vehicle meshes [85]. Such indirect incorporation of physical simulations, however, results in suboptimal efficiency and accuracy. Furthermore, due to the low expressibility of the backbone latent representation, the versatility of the generated results is very limited compared to current state-of-the-art diffusion-based models, as the results are typically confined to specific categories (e.g., furniture and vehicles). Most recently, Ni et al. bridged differentiable physical simulation with differentiable rendering to obtain virtual 3D reconstructions from real-world images that are physically plausible in simulators. Their work primarily focused on simple four-leg-supported objects such as tables and chairs [54]. Moreover, the evaluations of all aforementioned works are conducted in virtual simulators, leaving their performance in the real world untested. This limitation hinders potential downstream applications such as industrial manufacturing and robotic manipulation.

Since the pioneering work DreamFusion [58], Score Distillation Sampling (SDS) has demonstrated efficacy in elevating 2D content to 3D, inspiring numerous follow-up studies [8; 35; 46; 69; 88; 109]. These advancements have enhanced both the versatility of generated content and the quality of textures. However, none have addressed the crucial issue of physical stability. On the other hand, traditional computational fabrication has concentrated on employing topology and shape optimization to ensure that 3D printed objects can stand in a balanced state [59]. Directly integrating these methods with 3D generative AI as a postprocessing module is suboptimal. Shape optimization disregards the original input conditions of diffusion models, while topology optimization produces internal structures that defy intuitive physics, rendering them unsuitable for training embodied AI systems designed to emulate human-like reasoning about physical objects.

Observing this gap, we introduce Atlas3D, a generation pipeline that produces physically plausible, self-supporting 3D models from text. Incorporating differentiable physics-based simulation into our process, we generate models that are both simulation- and fabrication-ready. That is, they can be directly utilized in physical simulators, or 3D-printed for real-world applications; see Fig. 1 and Fig. 2. As our method is orthogonal to previous SDS-based techniques, which focus on non-physical qualities, it can be seamlessly integrated into many existing generation frameworks, functioning either as part of the refinement stage in a multi-stage method or as a post-processing step in a single-stage method. We demonstrate the efficacy of Atlas3D by comparing the stability of our models with those

produced by existing methods. Validation examples reveal that our generated models can be deployed as virtual simulation assets. Their stability transfers directly to the real world, as evidenced by our 3D printed results, suggesting further applications in robot training.

## 2 Related Work

**Diffusion-based 3D Generation**   Due to the abundance of information encoded in large image latent diffusion models (LDMs) [65], extensive studies have used pre-trained LDMs to distill 3D content. One approach is to fine-tune LDMs to support novel view synthesis, with a separate multiview fusion step to produce 3D content [39; 68; 99; 38; 37; 89; 95; 42; 80; 25; 106; 41; 9; 83]. Another approach, which is more related to our paper, is using LDMs as likelihood discriminators. A differentiable renderer is connected to a 3D representation, and the LDMs guide the optimization of the representation parameters. [58] proposed Score Distillation Sampling (SDS). Efficiency has been improved by coarse-to-fine strategies [35; 60; 79; 8] and timestep scheduling [18; 100]. 3D priors are involved to improve multiview consistency [108; 66; 31; 1; 93; 36; 78]. Multiview diffusions can also be used to evaluate SDS [69; 87; 105; 94]. Other researchers have explored SDS variants or improvements [84; 77; 88; 24; 101; 16; 109; 96]. 3D LDMs that directly generate 3D representations are also explored, such as compositional scenes [17; 57], point clouds [43; 81; 51], SDFs [70; 107; 30], occupancy fields [15; 72; 45; 12] and NeRFs [3; 11; 52; 6; 55; 5; 75; 23].

**Physics-aware 3D Generation**   Most existing 3D generative models focus only on geometry or appearance modeling, with physics priors being underexplored. Time-independent physical constraints, such as penetrations, can be directly defined by penalties [97; 17; 82; 40; 102]. For time-dependent physical qualities, such as stability and comfort, data-driven quality checkers trained with offline simulators can be applied [10; 48; 4]. Offline simulators can also be used as validators to augment the training dataset [71] and update the design with reinforcement learning [85], and as dynamics generators for generated 3D assets [91; 21; 62]. Another direction is to utilize differentiable simulations, which have be widely used in tasks like robotic control [74; 19; 61; 34; 76] or inverse problems [33; 20; 103; 53; 32; 73; 7; 104]. They can also be applied in 3D content generation to define physics-based losses to aid per-instance generation [54; 92] or model training [47].

## 3 Background

### 3.1 Score Distillation Sampling

SDS-based methods are shown to be effective in distilling 3D models from 2D images. They utilize a 3D representation such as implicit density field, implicit Signed Distance Field (SDF), or tetrahedral SDF [67], a differentiable renderer like NeRF [50], NeuS [86], or Nvdiffrast [26], and a pre-trained text-to-image model such as Stable Diffusion [64] serving as diffusion guidance. The generation process involves optimizing the parameters $\theta$ of the underlying 3D representation, where the 3D shape is differentiably rendered to 2D images $z = g(\theta)$ and compared against the real distribution from the diffusion model with text guidance $y$:

$$\mathcal{L}_{\text{SDS}} = \mathbb{E}_{t,\epsilon} \left[ w(t) \left\| \hat{\epsilon}_\phi(z_t, y, t) - \epsilon \right\|^2 \right],\tag{1}$$

where $z_t$ is the noisy image at noise level $t$, $w(t)$ is a weighting function, and $\hat{\epsilon}_\phi$ is the predicted noise. We refer readers to the Appendix for more technical details on SDS optimization.

### 3.2 Rigid Body Dynamics

To incorporate physics into our framework, we propose predicting the dynamics of the generated 3D models using a differentiable simulator, where all objects are treated as rigid bodies. We follow the conventions in [2] to define the dynamical states of the simulation.

The kinematics of a rigid body are described by its mass $M$ and body-space inertia tensor $\mathbf{I}_{\text{body}}$, which remains constant. Assuming the center of mass of the body initially lies at the origin, the physical state $\Psi$ of the body at time $t$ (not to be confused with the noise level in diffusion) includes position $\mathbf{T}(t)$ and orientation $\mathbf{R}(t)$ (spatial information), and its linear and angular momentum $\mathbf{P}(t)$ and $\mathbf{L}(t)$ (velocity information). The rigid body equations of motion are given by

$$\frac{d}{dt}\Psi(t) = \frac{d}{dt}\begin{pmatrix}\mathbf{T}(t)\\\mathbf{R}(t)\\\mathbf{P}(t)\\\mathbf{L}(t)\end{pmatrix} = \begin{pmatrix}\mathbf{v}(t)\\\boldsymbol{\omega}(t)*\mathbf{R}(t)\\\mathbf{F}(t)\\\boldsymbol{\tau}(t)\end{pmatrix}, \qquad (2)$$

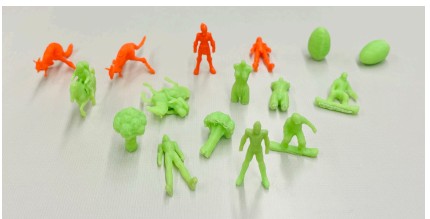

where $\mathbf{F}(t)$ and $\boldsymbol{\tau}(t)$ are the total force and torque exerted on the body, $\mathbf{v}(t) = \frac{\mathbf{P}(t)}{M}$ is the linear velocity, $\boldsymbol{\omega}(t) = \mathbf{I}(t)^{-1}\mathbf{L}(t)$ is the angular velocity, $\mathbf{I}(t) = \mathbf{R}(t)\mathbf{I}_{\text{body}}\mathbf{R}(t)^T$ is the world-space inertia tensor, and $*$ denotes cross product of $\boldsymbol{\omega}$ with the columns of $\mathbf{R}$. The physical state at a later time can be derived via time inte-

Figure 2: 3D-printed figurines created with Atlas3D stand stably, while those without Atlas3D have fallen down.

gration: $\Psi(t) = \Psi(0) + \int_0^t \frac{d\Psi}{ds}(s)ds$, which can be solved by numerical methods. By optimizing the physical states together with the SDS loss, we can jointly refine both the 3D geometry and the physical attributes of the generated results.

## 4 Atlas3D Algorithm

We introduce Atlas3D, a plug-and-play algorithm for generating 3D models from text. Focusing on man-made objects such as action figures and toys, which generally do not deform, Atlas3D treats generated models as rigid bodies and incorporates physics-based guidance into the generation process.

### 4.1 Physics Incorporation

As mentioned in § 3.2, we predict the dynamic behavior of generated models by rigid body simulations. While various explicit or implicit representations of 3D shapes can be chosen in a generation network, we opt for triangular meshes in our framework as they facilitate frictional contact modeling and simplify kinematics computation. Given a triangle surface mesh representation $\mathbf{X}(\theta)$, where $\theta$ is the implicit parameter, we integrate $\mathbf{X}$ into a rigid body represented by the dynamic state $\Psi(t) = [\mathbf{T}(t), \mathbf{R}(t), \mathbf{P}(t), \mathbf{L}(t)]^T$, where the world-space location $\mathbf{x}$ of any point $\mathbf{X}$ on the body is $\mathbf{x}(t) = \mathbf{R}(t)\mathbf{X} + \mathbf{T}(t)$. Assuming the 3D model is initially placed upright[3] on the ground with the bottom point touching the surface, we define standability as:

$$\lim_{t\to\infty}\Psi(t) = \Psi(0). \qquad (3)$$

Standability intuitively indicates an equilibrium state where all external forces acting on the object are balanced, and the physical state remains unchanged over time. However, perfectly placing an object straight on the ground without initial velocity is impractical in the real world. For example, when manually placing a cube on a flat table, the bottom face is unlikely to be perfectly parallel to the table surface. A stable 3D model should recover its initial state under mild perturbations, such as minor shaking. This state is known as stable equilibrium. Motivated by this, we augment standability with stable equilibrium $\tilde{\Psi}(t)$ defined as

$$\tilde{\Psi}(t) = \Psi(0) + \epsilon_0 + \int_0^t \frac{d\Psi}{ds}(s)ds \quad\text{and}\quad \lim_{t\to\infty}\tilde{\Psi}(t) = \Psi(0), \qquad (4)$$

where $\epsilon_0$ represents mild perturbations to the initial physical state. We first describe how to incorporate the standability criterion (Eq. 3) into the optimization process of 3D generation, and then explain how to further augment it with stable equilibrium (Eq. 4). Additionally, we introduce geometry regularization to enhance the smoothness of generated meshes.

### 4.1.1 Standability through Differentiable Simulation

We utilize a differentiable rigid-body simulator to obtain the physical state $\Psi(t)$. Assuming that the 3D model will eventually reach its steady state, $\exists T \in \mathbb{R}$ large enough such that $\forall t > T, \Psi(t) = \Psi(T)$. Let $\mathcal{S}$ denote a differentiable simulation function. We approximate $\Psi(T)$ via simulation:

$$\Psi(T) = \mathcal{S}(\mathbf{X}(\theta), \Psi(0), \mu, T). \qquad (5)$$

---

[3]We define the upright pose by setting the upward axis to coincide with the upward axis from the pre-trained text-to-3D model, as the upright direction is semantics-driven and generative models inherently learn these semantics from the training data.

Here $T$ is the simulation end time, $\mu$ captures material parameters such as density and friction coefficient, as well as simulator parameters such as time step and damping. We adopt the semi-implicit Euler time-integrator in Warp [44] for simulation.

Assuming the initial translation, rotation, and velocity are all zero, the difference between $\Psi(T)$ and $\Psi(0)$ arises only from discrepancies in spatial location, as the velocity at the final steady state is also zero. Therefore, we propose a standability loss to penalize rotational deviation due to instability:

$$\mathcal{L}_{\text{stand}} = \|\mathbf{R}(T) - \mathbf{R}(0)\|_2^2 = \|\mathbf{R}(T) - \mathbf{I}\|_2^2, \tag{6}$$

where $\mathbf{R} \in \mathbb{R}^{3 \times 3}$ represents the rotation matrix. We disregard the translation $\mathbf{T}$, as real-world instability mostly leads to rotational deviation from the initial state, such as falling to one side, while most translations, like falling due to gravity, are irrelevant to standability.

With a differentiable simulator, the standability loss can be backpropagated to mesh vertex coordinates and then to the implicit parameter $\theta$ as $\frac{d\mathcal{L}_{\text{stand}}}{d\theta} = \frac{d\mathcal{L}_{\text{stand}}}{d\mathbf{X}} \frac{d\mathbf{X}}{d\theta}$. In theory, any differentiable simulator is compatible with our framework.

### 4.1.2 Stable Equilibrium

Although the standability loss directly penalizes the non-standability of a 3D object, it can be slow to compute, especially when $T$ is large and many time steps are required, creating a huge computational graph. Consequently, both the simulation itself and the backpropagation of gradients through the simulation trajectory are time-consuming. Additionally, standability does not necessarily imply stable equilibrium, which is crucial for real-world 3D objects such as action figures and toys. Without this property, an object remains unstable even if standability is achieved, known as unstable equilibrium. Unstable equilibrium means that when a disturbance force is applied, the object moves away from its original position instead of recovering. Fig. 3 visualizes the difference between stable and unstable equilibrium. In the absence of perturbation, geometries like the upside-down triangle may remain standable in a simulator but are clearly unstable in the real world. Thus, we augment standability with stable equilibrium (Eq. 4). One straightforward way to incorporate this property is to introduce initial perturbation $\epsilon_0$ into the simulator. However, this would require many more simulations with various perturbations and subsequent loss backpropagation, which is extremely time-consuming.

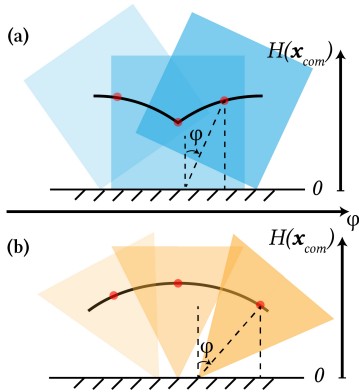

Figure 3: 2D illustration of stable equilibrium and unstable equilibrium. (a) A square is stable as a small perturbation of $\phi$ increases in $H(\mathbf{x}_{\text{com}})$;(b) An upside-down triangle is unstable as tilting decreases $H(\mathbf{x}_{\text{com}})$.

Inspired by the concept of a potential well [13], we augment our optimization objective with a robust and efficient stable equilibrium loss $\mathcal{L}_{\text{stable}}$. Specifically, for an object to be robustly standable, it needs to reside at a local minimum of potential energy—specifically, gravitational potential energy—so that if perturbed, gravity will act as a restoring force that returns the object to its original state. For a rigid body, gravitational potential energy is determined by the height of its center of mass. Thus, for any object in a stable equilibrium state, the center of mass would rise if it is slightly perturbed. This leads to our formulation of the stable equilibrium loss $\mathcal{L}_{\text{stable}}$. Let $\mathbf{x}_{\text{com}}$ denote the position of the center of mass of the underlying geometry and $H(\mathbf{x})$ denote the distance of the point $\mathbf{x}$ to the ground, assuming the object's pivot point is at $z = 0$. The stable equilibrium loss is defined as:

$$\mathcal{L}_{\text{stable}} := \mathbb{E}_{\mathbf{v} \in \mathbb{R}^2, \|\mathbf{v}\|=1} \left[ \max\{H(\mathbf{x}_{\text{com}}(\mathbf{P}_{\mathbf{v}}^{\phi}\mathbf{X})) - H(\mathbf{x}_{\text{com}}(\mathbf{X})), 0\} \right], \tag{7}$$

where $\mathbf{P}_{\mathbf{v}}^{\phi}$ represents the rotation of $\phi$ radian about axis $[\mathbf{v}^T, 0]^T$. Mathematically, a local minimum of gravitational potential energy is reached if $\exists \phi_0$ such that $\forall \phi \in (0, \phi_0), \mathcal{L}_{\text{stable}} = 0$. In practice, we fix the perturbation scale $\phi$ and uniformly sample 20 perturbation directions $\mathbf{v}$ in $xy$-plane.

### 4.2 Additional Regularization

While standability loss $\mathcal{L}_{\text{stand}}$ and stable equilibrium loss $\mathcal{L}_{\text{stable}}$ provide a well-defined objective for robust standing, they may lead to distorted optimized meshes due to the high-dimensional searching

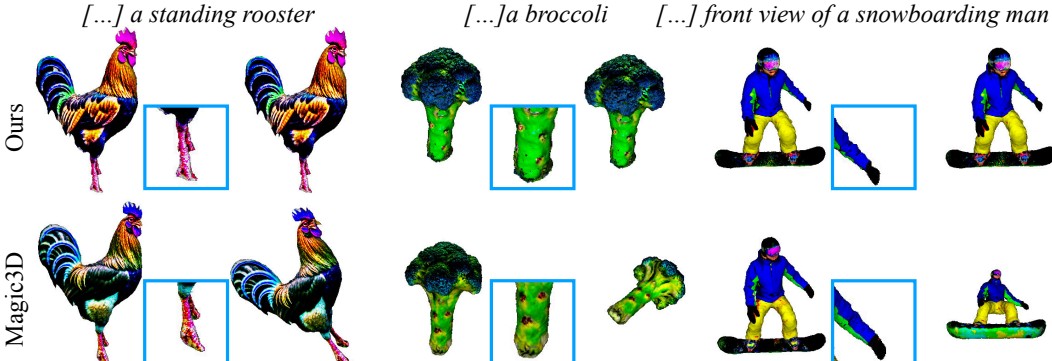

Figure 4: Comparison with Magic3D [35] includes zoom-in views that highlight the detailed changes in geometry. Our method enhances Magic3D with physics priors to generate self-supporting meshes.

space of implicit parameter $\theta$ without constraint. To constrain the optimization space and obtain smooth meshes, we add a normal consistency term that favors smooth solutions:

$$\mathcal{L}_{\text{normal}} = \frac{1}{|\mathcal{T}|} \sum_{(i,j)\in\mathcal{T}} (1 - \mathbf{n}_i \cdot \mathbf{n}_j), \tag{8}$$

where $\mathcal{T}$ is the set of the triangle pairs sharing a common pair with $\mathbf{n}_i, \mathbf{n}_j$ being their normals respectively. This term maximizes the cosine similarity between neighboring surface triangle normals, leading to smoother meshes. Considering the bottom surfaces of most robust standing objects are flat, we apply the Laplacian loss to a subset $\mathcal{B}$ of vertices with a height lower than a threshold $h_b$:

$$\mathcal{L}_{\text{b-lap}} = \frac{1}{|\mathcal{B}|} \sum_{i\in\mathcal{B}} \|\boldsymbol{\delta}_i\|_2, \tag{9}$$

where $\boldsymbol{\delta}_i = (\mathbf{L}\mathbf{V})_i \in \mathbb{R}^3$ calculates the differential coordinates of vertex $i$ with $\mathbf{L}$ being the Laplacian matrix of the mesh graph and $\mathbf{V}$ representing mesh vertices. Intuitively, this loss term attempts to minimize the distance between vertex $i$ and the average position of adjacent vertices.

### 4.3 Method Overview

With the physically-inspired loss terms derived above, we now describe how to incorporate them into the text-to-3D framework. SDS-based methods and their variants start optimization with a random initialization of the implicit parameters, which initially have no knowledge of the model's geometry. Adding physical constraints at this early stage would be ineffective. Therefore, we propose a two-stage training strategy: the coarse stage and the refine stage. In the coarse stage, we generate a rough shape of the model using a text prompt. We can adopt any SDS-based generation framework as our baseline model, offering various choices of implicit representation and differentiable renderers. In the refine stage, we optimize the geometry with our physical constraints included. For this, we use a tetrahedral SDF representation [67] and employ Deep Marching Tetrahedra (DmTet) to differentiably convert the coarse geometry from implicit density or SDF as necessary. We utilize Nvdiffrast [26] as the differentiable renderer and Stable Diffusion v2.1 [64] for guidance. We propose the following loss function for the joint optimization of texture, geometry, and stability:

$$\mathcal{L} = \lambda_{\text{SDS}}\mathcal{L}_{\text{SDS}} + \lambda_{\text{stand}}\mathcal{L}_{\text{stand}} + \lambda_{\text{stable}}\mathcal{L}_{\text{stable}} + \lambda_{\text{normal}}\mathcal{L}_{\text{normal}} + \lambda_{\text{b-lap}}\mathcal{L}_{\text{b-lap}} \tag{10}$$

In practice, we observe that adding standability loss once every 10 iterations is sufficient to ensure a significant reduction in loss without notably increasing computational overhead.

## 5 Experiments

In this section, we devise comprehensive experiments (both virtual and real-world) to demonstrate the efficacy of our method. We use a series of text prompts to generate 3D models that we expect to be self-supporting, and compare the generated results with baseline models. Our models are verified by simulation for stability and then fabricated using a 3D printer for real-world testing. We refer readers to the Appendix for more details about implementation, training, and experiment setup.

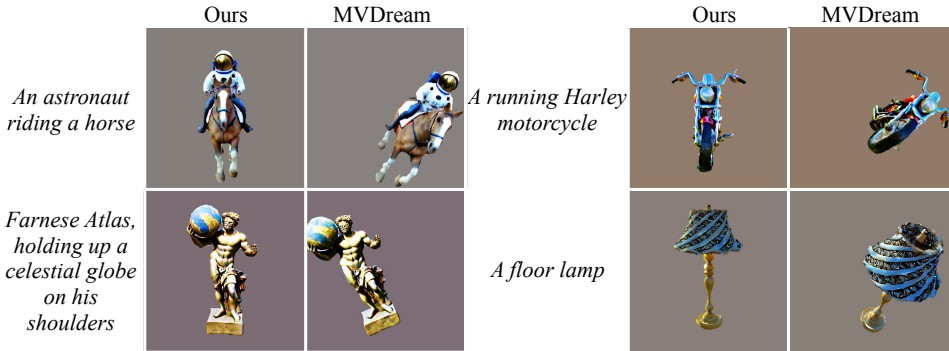

|  | Ours | MVDream |  | Ours | MVDream |
|--|------|---------|--|------|---------|
| *An astronaut riding a horse* | | | *A running Harley motorcycle* | | |
| *Farnese Atlas, holding up a celestial globe on his shoulders* | | | *A floor lamp* | | |

Figure 5: Atlas3D is also compatible with MVDream [69], enhancing it with stable standability.

## 5.1 Simulation Verification

**Qualitative Comparison**    Using the same text prompts, we compare our generated models with previous methods. The quality of the results is assessed by their stability, which is verified by a forward simulation: we simulate the generated models in an upright initial position close to the ground for a sufficiently long time and record whether they fall. Using Magic3D [35] as the baseline model, we visualize the initial state and a later state in Fig. 4. Our generated meshes remain stable throughout the simulation, while the baseline models fail under the same conditions due to a lack of consideration for physics.

We highlight the main changes in mesh topology that enable the models to stand (see Fig. 4). These changes include modifications to both the overall shape and specific local geometries. More specifically, our physical adjustments alter the contact surface to gain more support from the ground and shift the center of mass to be slightly lower and more centered above the contact surface. These macroscopic and microscopic optimizations jointly increase the support to the models, thus ensuring their stability. Note that our method slightly modifies the texture of the generated results as we are jointly optimizing our physical adjustments with the SDS loss.

Since we do not assume a specific baseline model in the first stage, we can vary the model used in the coarse stage to generate versatile, physically-aware 3D meshes based on different existing SDS generation models. We use MVDream [69] as another baseline model and compare the results side-by-side with ours in Fig. 5. Our method improves the geometry of the mesh and ensures stability in simulation. The quality of the texture and the main part of the shape is determined by the underlying model used in the coarse stage, while we focus on improving the physical stability in the refinement stage. More qualitative comparisons with previous methods are provided in the Appendix.

**Ablation Study**    We perform ablation studies to demonstrate the necessity of our proposed losses. It can be observed in Fig. 6(b), that without standability loss $L_{\text{stand}}$, the model fails to stand. While the model can still stand without stable equilibrium loss $L_{\text{stable}}$, as demonstrated in Fig. 6(c), it is less stable under perturbation (see next section for details). The geometry regularization loss term $L_{\text{b-lap}}$ helps smooth the geometry and avoid spiky artifacts on the surface as shown in Fig. 6(d). Additionally, we show that applying mesh regularization as a post-processing step, rather than integrating it with SDS loss in a joint optimization, can degrade text alignment as this neglects the semantics during the deformation process (see Fig. 6(e)).

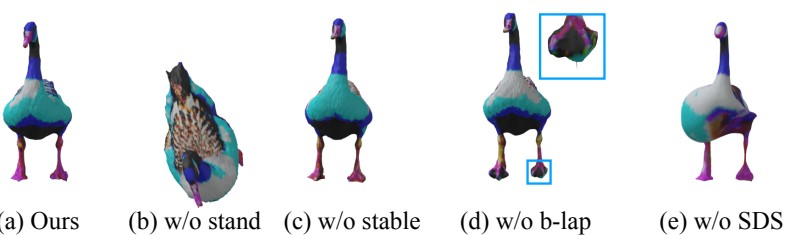

(a) Ours    (b) w/o stand    (c) w/o stable    (d) w/o b-lap    (e) w/o SDS

Figure 6: Ablation study of each loss term.

Table 1: Comparison of success rate under perturbation (goose).

| Perturbation Angle $\theta_{\max}$ | 0 | 0.01 | 0.02 | 0.04 | 0.08 |
|---|---|---|---|---|---|
| w/ stability loss | 1 | 1 | 0.99 | 0.69 | 0.4 |
| w/o stability loss | 1 | 0.97 | 0.71 | 0.62 | 0.23 |

**Stability under Perturbation** In the real world, placing an object on the ground always involves some noise, as both human and robot manipulation have imprecision in angles and directions. Hence, the standability of an object under small perturbations is crucial for improving the success rate of such tasks.

To mimic this uncertainty in our framework, we evaluate the stability of our generated models under a small initial rotation. With a given precision $\phi_{\max}$, we rotate the generated mesh at random angles $\phi_y, \phi_x \in (-\phi_{\max}, \phi_{\max})$ in both the $y$ and $x$ axes, respectively (with the $z$ axis being the up direction). The mesh is then placed close to the ground and tested in a simulation to see if it can still stand. We choose 13 different values of the maximum perturbation angle $\phi_{\max}$ and perform 100 random tests with each angle on 6 of our generated models. We report the success rate in Fig. 7. We define a successful test as: after a sufficiently long time period, the maximum height of the model stays within 3% of the initial maximum height.

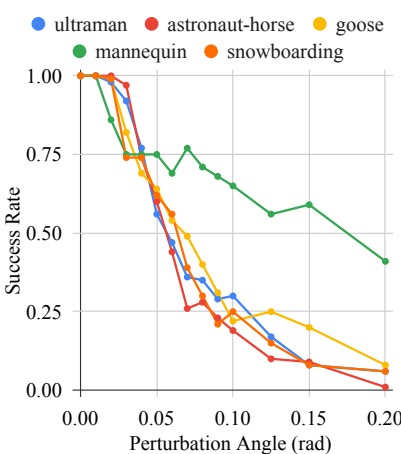

Figure 7: Success rate of models standing under perturbation.

Due to the presence of physical constraints, our generated models can withstand small initial perturbations, while the baseline models fail to stand when placed straight up (without rotation), let alone with perturbations. Furthermore, introducing the stable equilibrium loss consistently increases the success rate of standing under different scales of perturbations, as shown in Table. 1.

**Standability on Different Platforms** Our pipeline can be generalized to learn standability on various platforms, not just flat ground, by incorporating them as boundary conditions in the simulator. To demonstrate this, we use a 10-degree inclined plane and a sphere, then train our 3D model separately to stand still on each. Modeling frictional contact is crucial for achieving stability in such scenarios. As shown in Figure Fig. 8, our optimized mesh stands stably on both the incline and the sphere with the help of static friction, whereas the baseline model fails as expected.

**Simulator Cross-validation** While we base our method on a differentiable rigid-body simulator with a semi-implicit Euler time-integrator, our pipeline is compatible with any other physics-based simulator as the backbone, with differentiability required for the training stage. To verify the reliability of models generated with our simulator, we include an external simulator in the testing stage to verify the correctness of the simulated dynamics. We choose the Incremental Potential Contact (IPC) method [27; 29], which has been proven accurate for frictional contact. We validate the correctness of every single generated model and visualize the simulation results in Fig. 1.

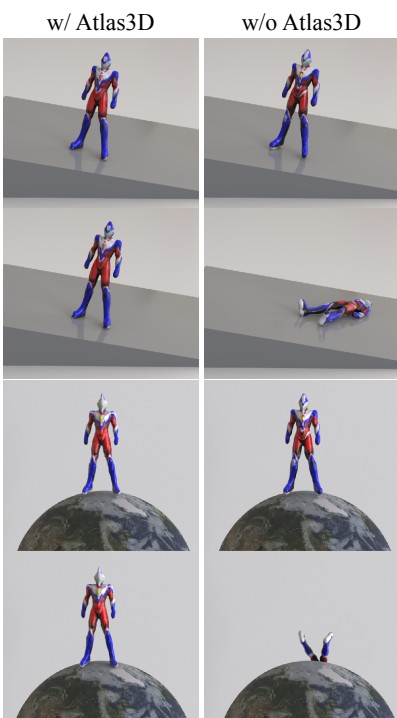

Figure 8: Standability evaluation on uneven surfaces.

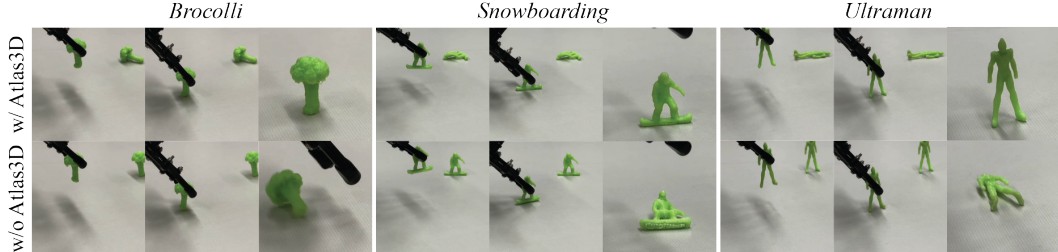

*Brocolli*       *Snowboarding*       *Ultraman*

w/ Atlas3D   w/o Atlas3D

Figure 10: Standability test using a robotic arm. More results are shown in the Appendix.

**Quantitative Evaluation** We examine the versatility of our method. We randomly select 150 prompts from [58] and manually exclude 43 prompts deemed unfeasible (for instance, it does not make sense to require "a swan and its cygnets swimming in a pond" to be standable), leaving a total of 107 prompts. We use the two-stage Magic3D [35] as the baseline model and compare our optimized mesh with the results from the refine stage of the baseline under the same settings (e.g., iterations, loss weights).

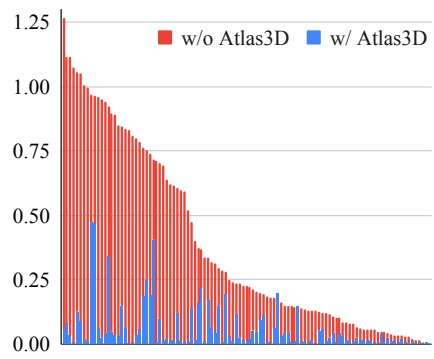

Figure 9: TRD results from 107 prompts using the Magic3D baseline and our method.

To evaluate the standability of the baseline method and our method, we run the rigid body simulation in Warp with simulation end time $T = 2.0$ at which almost all objects have reached the steady state. We propose Time-Averaged Rotation Deviation Loss (TRD) defined as

$$\text{TRD} = \frac{1}{T} \int_0^T ||\mathbf{R}(t)\hat{\mathbf{z}} - \mathbf{R}(0)\hat{\mathbf{z}}||_2 dt \qquad (11)$$

to assess the standability, as a representation of the average tilting of the upward direction $\hat{\mathbf{z}}$ (of the object) over time. We approximate the integral (Eq. 11) with discrete quadrature $\Delta t = 0.02$. Results are plotted in Fig. 9 [4]. The mean TRD score is reduced by more than six times compared with the baseline method. As shown in Table. 2, we calculate the average CLIP score [63] of 107 generated shapes for both our proposed method and baseline. Furthermore, Elo (GPT-4o) [90] scores are presented. Both metrics illustrate that our method not only implements physics-based stability but also maintains the fidelity of the generated 3D shapes in terms of content alignment and overall shape quality. More details are provided in the Appendix.

## 5.2 Real-world Validation

One major advantage of incorporating physics-based simulation into the optimization pipeline is that it bridges the gap between the generated model and the real world. Our method ensures the direct usability of the model for fabrication, with success primarily dependent on the accuracy of both the simulator and the manufacturing machine.

Table 2: Quantitative Evaluation

| Metrics | Ours | Magic3D |
|---|---|---|
| TRD ↓ | 0.060 | 0.389 |
| CLIP ↑ | 25.356 | 25.781 |
| Elo (GPT-4o) ↑ | 970.774 | 1029.226 |

**3D Printing and User Studies** We test the readiness of our generated meshes for real-world application by producing eight figures using a 3D-printing device (Zortrax M200 with Z-ABS filament material). For reference, we also print the corresponding baseline meshes generated without physical constraints. Our physically constrained figures can steadily support themselves when gently placed on an even surface, while the baseline figures either fail to stand at all, or require extensive adjustments and fall easily with little perturbation (see Fig. 2).

We conduct user studies with these printed figures to assess their stability under different types of human manipulation. Ten users were asked to place the figures upright on a table, with five trials for each figure, resulting in a total of 800 trials. Fabricated figures generated from the baseline has a success rate of 7%, while figures generated from our method has an overall success rate of 92.25%.

---

[4]Results are ranked by TRD scores of the baseline approach.

Itemized results are provided in the Appendix. Our method significantly increases the physical stability of the models under varying human efforts.

**Validation with a Robot Arm**   To showcase the compatibility of our framework with robotic applications, we test our fabricated figures with a teleoperated LewanSoul LeArm robot arm outfitted with a two-finger parallel gripper (see Fig. 10). The gripper is set to initially grasp a figure above the ground. It slowly moves downward, and is then gently released to place the figure on the ground. Four trials were performed for each figure, yielding 64 trials in total. For the baseline method, 6.25% trials resulted in successful standing figures, while ours has a success rate of 90.6%. Experimental data are provided in the Appendix and supplemental video.

## 6   Conclusion

We present Atlas3D, a physically constrained SDS-based framework that generates self-supporting 3D models from text prompts. Our framework can learn standability through a differentiable physics-based simulator and other physics-inspired loss functions. The generated 3D models can be directly imported into a physics simulator and are ready to be manufactured and deployed in the real world. Our method has wide potential for generation tasks, as it can be easily integrated into many existing pipelines and improve the physical plausibility of their generated results.

**Limitations and Future Work**   Our physical adjustments are optimized over all mesh vertices, resulting in a large degree of freedom in optimization. This may lead to undesired distorted meshes [49]. Future works may consider adding a latent embedding or skeleton rigging to limit the variety of mesh deformation. Our framework focuses on SDS-based methods as a backbone. It would be interesting to further generalize our physical constraints to other non-SDS or non-diffusion based methods [22; 49; 28]. Finally, we only consider text-to-3D tasks in this work. An exciting extension is to generalize our work to image-to-3D tasks [60; 39].

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

# A  Appendix

## A.1  Score Distillation Sampling

Score Distillation Sampling (SDS) has proven to be an effective method for distilling 3D models from 2D images. SDS-based methods leverage a combination of 3D representations, differentiable rendering, and pre-trained text-to-image diffusion models to optimize 3D shapes with high fidelity and realism.

The generation process in SDS methods involves optimizing the parameters $\theta$ of the 3D representation. At each iteration, the 3D geometry is differentiably rendered into a 2D image $z = g(\theta)$. This image is then compared with the distribution of real images as modeled by the diffusion model. More specifically, the input image $z$ is first noised with a random noise $\epsilon$ at a specified noise level $t$. The diffusion model then predicts the noise $\hat{\epsilon}_\phi$ with text guidance $y$, and this prediction is compared with $\epsilon$, resulting in the following loss:

$$\mathcal{L}_{\text{SDS}} = \mathbb{E}_{t,\epsilon} \left[ w(t) \, \| \hat{\epsilon}_\phi(z_t, y, t) - \epsilon \|^2 \right], \tag{12}$$

where $w(t)$ is a weighting function modulating the influence of different noise levels. The expectation is taken over the noise level $t$ and noise term $\epsilon$, ensuring that the generated 3D shape aligns with the text-guided distribution of images.

The gradient with respect to the optimization parameter $\theta$ is then backpropagated as follows:

$$\nabla \mathcal{L}_{\text{SDS}}(z = g(\theta)) = \mathbb{E}_{t,\epsilon} \left[ w(t) \left( \hat{\epsilon}_\phi(z_t, y, t) - \epsilon \right) \frac{\partial z}{\partial \theta} \right], \tag{13}$$

where the U-Net Jacobian term $\frac{\partial \hat{\epsilon}_\phi}{\partial z_t}$ is omitted for efficient optimization.

For further technical details on SDS optimization, we refer readers to [58; 35; 46].

## A.2  Implementation and Training Details

We implement our pipeline in PyTorch with Adam optimizer. For differentiable simulation, we adopt the semi-implicit Euler simulator in Warp [44], where gradients of physical states are computed via auto-differentiation and backpropagated to the parameters of 3D representations [44; 56]. We use a two-stage training strategy and leave the choice of the first stage open for various SDS-based methods as baselines. For a two-stage baseline method, we implement our physics-inspired losses as submodules that can be seamlessly integrated into the refinement stage. For a one-stage method, our approach can be used as a standalone refinement stage to improve the physical quality of the baseline. For baseline methods that are not publicly available, we use the reimplementation from threestudio [14].

We train our models using a single NVIDIA RTX 3090 GPU. During our refinement stage, we implement a skipping strategy, incorporating standability loss only once every 10 iterations. In our quantitative evaluation of a batch of prompts, we observed an average refinement time of 36 minutes for each training step, with a default setting of 5,000 iterations.

For the rigid body simulator in Warp, we set $dt = 10^{-3}$s. Contact stiffness and damping are set to $10^3$ and 2.0; friction coefficient is set to 0.5; stiffness of friction force is set to $10^3$. Density of the 3D objects is set to $10^3$.

In our experiments, we use the following default weights for the loss terms: $\{\lambda_{\text{SDS}} = 1, \, \lambda_{\text{normal}} = 10^4, \, \lambda_{\text{stand}} = 10^5, \, \lambda_{\text{stable}} = 10^5, \, \lambda_{\text{b-lap}} = 10^7\}$. For some examples, we tune these weights within the following ranges: $\{\lambda_{\text{stand}} = 10^5 \sim 5 \times 10^5, \, \lambda_{\text{stable}} = 10^5 \sim 5 \times 10^5, \, \lambda_{\text{b-lap}} = 10^6 \sim 10^7\}$. Our heuristic intuition is to keep the SDS and physical loss terms roughly on the same scale. For the regularization terms, we scale them to around $1/1000$ to $1/100$ of the SDS and physical loss terms.

### A.3 Comparison with Post-processing Methods

While directly post-processing 3D generated models is a straightforward and effective approach to achieving physical stability, it may result in undesirable outcomes such as misalignment with the text prompt, as it overlooks semantics.

One simple post-processing method is to cut the mesh by a flat plane slightly higher than the lowest vertex. However, this method will fail when the projection of the center of mass lies outside the contact region, as shown in Fig. 11. Additionally, determining the cutting height is another parameter that must be manually set or optimized, and this may also degrade the overall appearance.

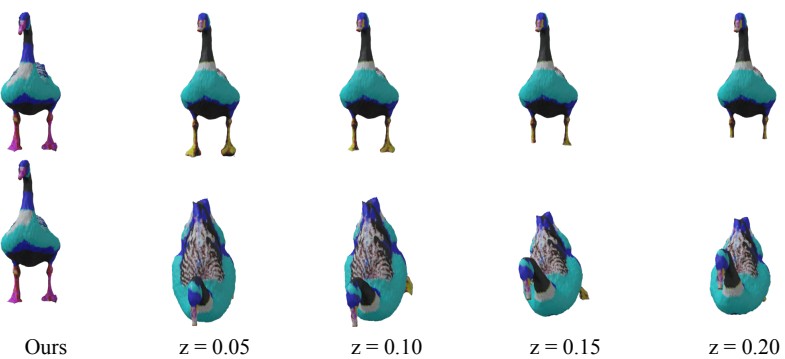

| Ours | z = 0.05 | z = 0.10 | z = 0.15 | z = 0.20 |

Figure 11: Comparison with cutting the mesh by a flat plane at height $z$.

Another post-processing method, *make-it-stand* [59], offers an effective way to relocate the center of mass to achieve standability. However, it assumes that the supporting surface is fixed during optimization, which can lead to distorted results due to the imperfect quality of text-to-3D generated models. We provide two examples in Fig. 12. For the goose example generated by Magic3D, one leg is shorter than the other. *Make-it-stand* only treats one foot as the supporting surface and ignores the other due to its post-processing nature, whereas our joint-optimization pipeline enables stable standing with two legs on the ground. A similar issue also happens to the kangaroo example. More importantly, the text alignment degrades as semantics are overlooked during optimization. In contrast, our proposed joint optimization method preserves the text alignment and dynamically adjusts the center of mass as well as the supporting surface configuration.

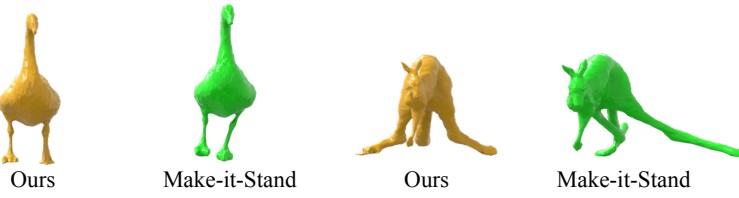

| Ours | Make-it-Stand | Ours | Make-it-Stand |

Figure 12: Comparison with *make-it-stand*.[5]

In Fig. 6(e) in the main text, we also apply our proposed losses in a post-processing manner. Similarly, while it is able to optimize the 3D models to make it standable, the text alignment is compromised. Overall, compared to the post-processing method, joint optimization is a more robust way which better balances text alignment and physical constraints.

---

[5]Although the output mesh of *make-it-stand* has the same resolution as the input mesh, the ordering of vertices and surfaces are shuffled in their implementation. As a result, we are unable to re-attach the original textures onto the output mesh. We are therefore unable to run metrics like CLIP in the paper to compare text alignment. Nevertheless, it should be visually clear that our results do have better text alignment.

### A.4 Additional Results

### A.4.1 Qualitative Comparison with Magic3D Baseline

Ours                                                                 Magic3D

*[...] a standing goose*

*[...] a standing pigeon*

*[...] a jumping kangaroo*

*[...] a mannequin*

*[...] a standing Konan*

*[...] a standing ultraman*

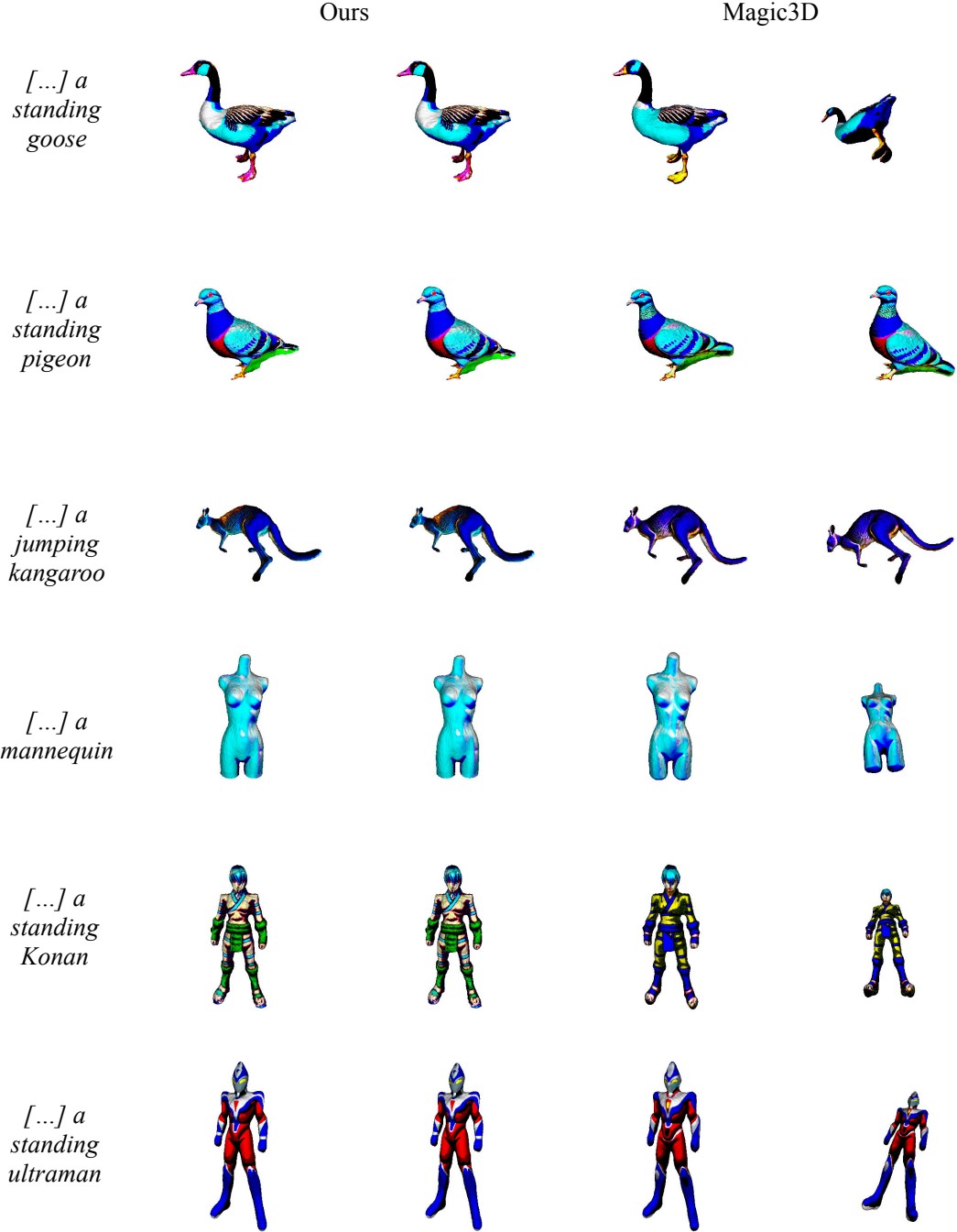

Figure 13: More comparison with Magic3D baseline

Ours                                    Magic3D

*[...] a toy
robot*

*[...] a robot
made out of
vegetables*

*[...] a goose
made out of
gold*

*[...] small
saguaro
cactus
planted in a
clay pot*

*[...] a baby
dragon
hatching out
of a stone
egg*

*[...] a bear
playing
electric bass*

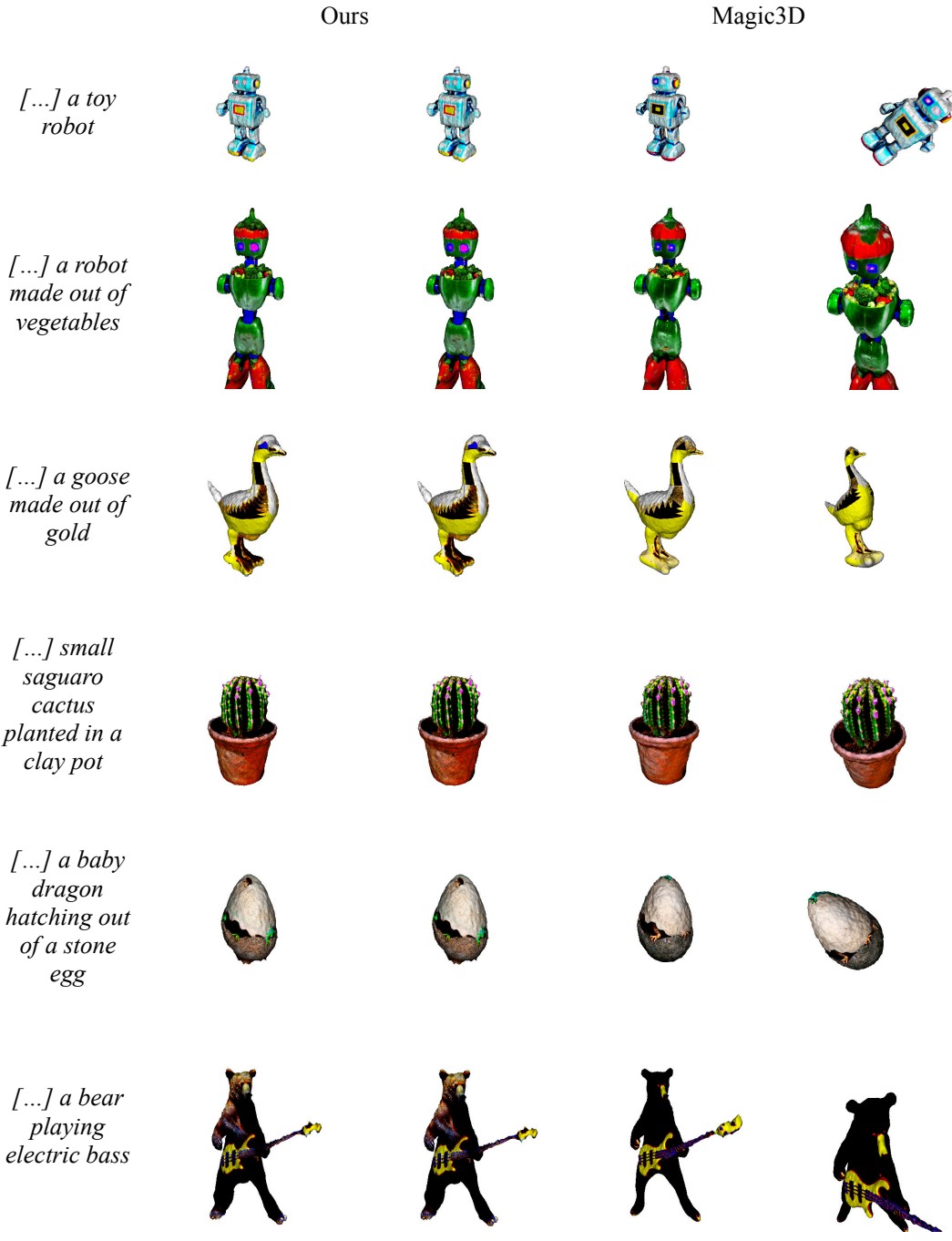

Figure 14: More comparison with Magic3D baseline

Ours                                          Magic3D

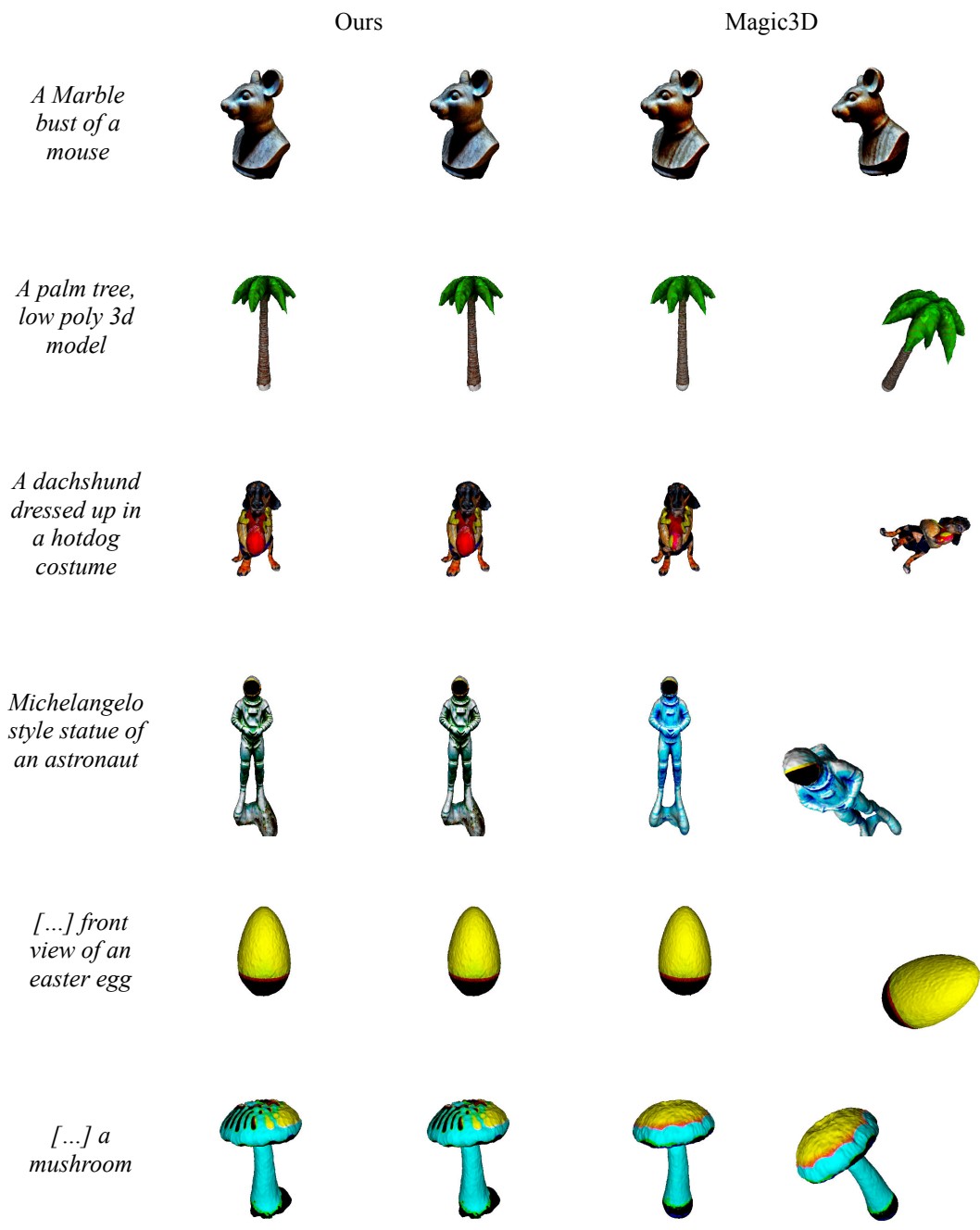

*A Marble bust of a mouse*

*A palm tree, low poly 3d model*

*A dachshund dressed up in a hotdog costume*

*Michelangelo style statue of an astronaut*

*[...] front view of an easter egg*

*[...] a mushroom*

Figure 15: More comparison with Magic3D baseline

### A.4.2 Qualitative Comparison with MVDream Baseline

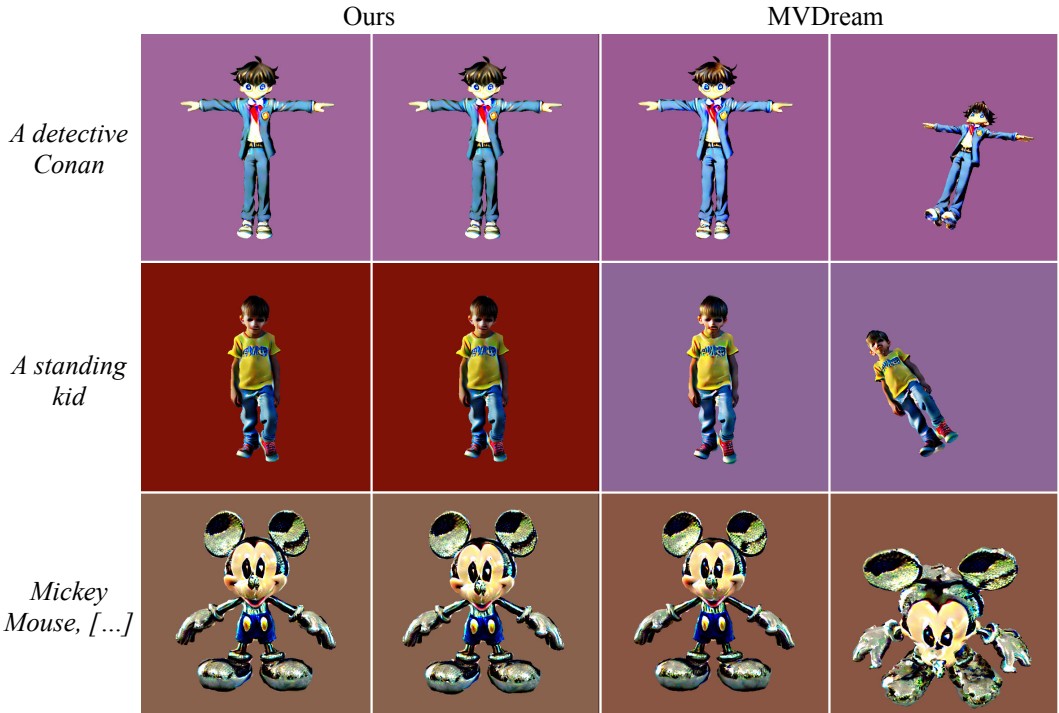

Figure 16: More comparison with MVDream baseline

### A.4.3 Mesh Topology Change

We provide a zoomed-in view of the local mesh topology change in Fig. 17 to demonstrate how our method optimizes the object's geometry to make it standable.

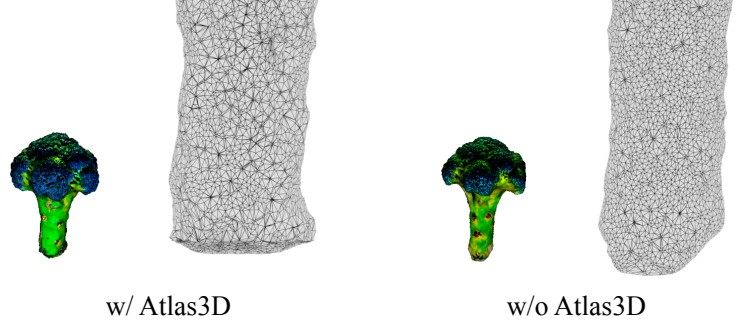

Figure 17: Local mesh topology change

### A.4.4 Quantitative Comparison with Magic3D Baseline

The results of the quantitative experiments are shown in 2. For CLIP score calculation, we specifically employ *openai/clip-vit-large-patch14-336* as the check point of CLIP model. We rendered images from various angles (0-360 degree, 3 degree as the interval). For Elo (GPT-4o) benchmark, we made use of the newly released GPT-4o model instead of the one in [90] due to capacity limitation. Additionally, we reduce the number of views to 6 because of token length limitation.

## A.5 Real-world Experiments

### A.5.1 Robot Manipulation Results[6]

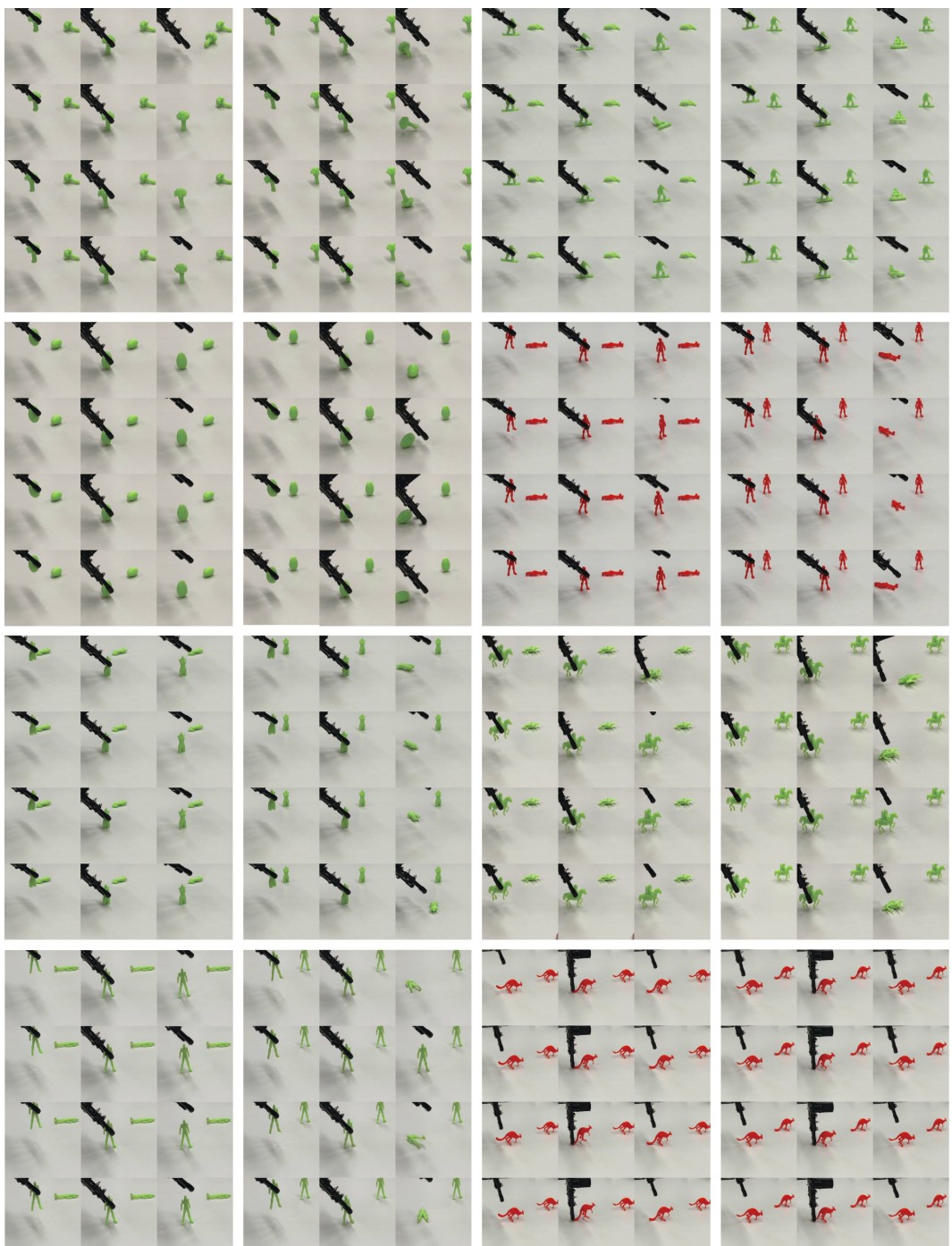

Figure 18: Robot manipulation experiment

We record our robot manipulation experiment in Fig. 18 and summarize quantitative results in Table. 3.

---

[6]We define as upward pose for the *kangaroo* figure as its tail touching the ground, as a real kangaroo does.

Table 3: Number of successes in robotic trials.

| Figure | Broccoli | Egg | Horse | Kangaroo | Konan | Mannequin | Snowboarding | Ultraman |
|---|---|---|---|---|---|---|---|---|
| Baseline | 0 | 0 | 1 | 0 | 0 | 0 | 0 | 1 |
| Ours | 3 | 4 | 3 | 4 | 4 | 4 | 3 | 4 |

### A.5.2 User Study Results

We report the detailed results of our user studies in Table. 4.

Table 4: Number of successes in user studies.

| Figure | Broccoli | Egg | Horse | Kangaroo | Konan | Mannequin | Snowboarding | Ultraman |
|---|---|---|---|---|---|---|---|---|
| Baseline | 0 | 0 | 8 | 4 | 6 | 2 | 1 | 7 |
| Ours | 44 | 46 | 40 | 48 | 47 | 50 | 45 | 49 |

