# OpenReview forum: "Atlas3D: Physically Constrained Self-Supporting Text-to-3D for Simulation and Fabrication"
_NeurIPS.cc/2024/Conference — NeurIPS 2024 poster_

### Official Review · Reviewer_iQck · 2024-07-11

**Soundness:** 2
**Presentation:** 3
**Contribution:** 2
**Rating:** 5
**Confidence:** 4

**Summary:**

This paper proposes to combine physical standability with existing diffusion-based text-to-3D generators in order to synthesize objects that not only follow the text description but also can stand on their own without falling. To this end, the paper proposes a physical loss function and uses it with score distillation sampling methods that generate 3D objects from texts. The paper shows that their generated model show better standability both in simulation and after physical realization.

**Strengths:**

- Good exposition and writing, paper is rather easy to follow.
- The validation of results especially the 3D printing of generated models.
- The standability and stable equilibrium losses are elegant.

**Weaknesses:**

First, there are good methods, such as 'make it stand' that could be used as post-processing tools for making text-based generated models stand. Why should we go through the paper's approach in the light of these methods?
Some important ablations are missing. For example,
- Separating of how much each loss is contributing to standability. For example, what if we use Magic 3D outputs and smoothen the geometry?
- What are the pros and cons of using the physical loss as a post-processing or jointly with SDS?


The claims are more broad than the actual contributions.
- Take "fabrication-ready" in line 62. The paper is concerned with stabdability and it should adapt the text (and even the title). In 3D printing, "self supporting" means that there is no need to support materials. The paper doesn't cover this.
- The relationship to robotics is not well-founded except an example that uses robot grippers to do the standability tests, which is not really
 related to benefits to robotics.

The paper is heavily relied on score distillation sampling methods but does not explain them very well. I understand that the paper refers to the original papers in this area but it is worth including a paragraph about SDS.

I look forward to see a justification in the rebuttal.

**Questions:**

- The paper mentions too many steps of integration and back propagation in Section 4.1.1. Does it still do this? If not, where exactly the differentiable physics simulation is used?
- What is the default intimal physical state?
- How does the paper figures out the upright position?
- The paper mentions mesh topology (Line 246) but doesn't show it. A zoom in mesh is necessary for this.
- What are the mesh sizes and dimensionalities of printed objects?
- Are Figure 7 results printed?
- x_{com} is not defined in Eq. 7.

**Limitations:**

A main limitation is the mesh size. The paper has mentioned it. This is important in the light of 3D printing that require sometimes huge resolutions.

---

> ### Author Rebuttal · Authors · 2024-08-05
>
> We thank you for recognizing the effectiveness and elegance of our pipeline. Below we clarify your questions.
>
> **Q1: First, there are good methods, such as 'make it stand' that could be used as post-processing tools[...]. Why should we go through the paper's approach in the light of these methods?**
>
> We provide additional comparisons with cutting the mesh and *make-it-stand* [59] in the attached PDF. In our global response, we discuss in detail the drawbacks of such post-processing methods compared to our joint optimization approach. Please refer to the global response for more details. Overall, post-processing methods overlook semantics during optimization, potentially leading to degraded text alignment. Also, *make-it-stand* [59] assumes the contact region is unchanged. Such inflexibility may result in undesired outcomes, as demonstrated in Fig. 2. In addition, To deform the underlying mesh for improving standability, *make-it-stand* [59] requires manually placing a few handles, whereas our method requires no human intervention and can thus be used for batch generation, as we did in Fig. 8 in the paper. Hence, we believe that our method is a streamlined plug-in tailored for 3D generative models.
>
> **Q2: Separating of how much each loss is contributing to standability.**
>
> Thank you for your suggestion! We have added more ablation studies to demonstrate the necessity of our proposed losses. Please refer to Fig. 3 and Table 1 in the attached PDF and the global rebuttal for more details. Additionally, we would like to point out that for all our comparisons with Magic3D outputs, we applied the same scale of normal smoothing to both our method and the baseline method.
>
> **Q3: What are the pros and cons of using the physical loss as a post-processing or jointly with SDS?**
>
> As stated in our response to Q1, post-processing methods overlook semantics during optimization, which can lead to degraded text alignment. We also applied our proposed physical loss in a post-processing manner, as shown in Fig. 3(e) in the attached PDF. While the generated results are still able to stand, the text alignment is significantly compromised.
>
> **Q4: The paper is concerned with standability and it should adapt the text (and even the title). In 3D printing, "self supporting" means that there is no need to support materials.**
>
> Thanks for pointing out this! Our definition of self supporting is different from the one in 3D printing. Specifically, we consider self supporting as the condition where, once the upright pose is determined by the text-to-3D model, the generated 3D model should satisfy the standability criteria defined in Eq. (3). We will clarify this in the revised version. In this work, we mainly use 3D printing as a verification in the real world settings.
>
> **Q5: The relationship to robotics is not well-founded except an example that uses robot grippers to do the standability tests [...].**
>
> The relevance of our work to robotics lies in our capability to automatically generate standable 3D assets. A simple scenario would involve training a robot to lift, move, and place objects without tipping them over. Our method ensures that the generated contents can stand stably, as is necessary for many real-world objects. We aim to promote the generation of 3D assets with automatic incorporation of physical standability, and we believe this property can effectively help robots practice accurate interaction with objects, such as grasping, picking, and placing, both in simulation and in the real world.
>
> **Q6: The paper is heavily relied on score distillation sampling methods but does not explain them very well. [...] it is worth including a paragraph about SDS.**
>
> We will revise Section 3.1 and include a more detailed description of SDS in the appendix.
>
> **Q7: The paper mentions too many steps of integration and back propagation in Section 4.1.1. Does it still do this? If not, where exactly the differentiable physics simulation is used?**
>
> In line 166, we mentioned that applying physical simulation at every optimization step can be time-consuming due to the many steps of forward time-stepping and backward propagation. In all our experiments, we apply the standability loss (physical simulation) once every 10 iterations, and we find that this approach is sufficient to ensure significant loss reduction and does not notably increase computational overhead (see line 226).
>
> **Q8: What is the default intimal physical state?**
>
> We treat the meshes as solid, uniform objects. We outlined the default physical parameters and simulation settings in Appendix A.1.
>
> **Q9: How does the paper figures out the upright position?**
>
> As elaborated in our global response, the upright direction is semantics-driven and generative models inherently learn these semantics from their training data. We thus define the upright pose by defining the upward axis to coincide with the upward axis from the default 3D-generated result. We will make this point clearer in the paper.
>
> **Q10: The paper mentions mesh topology (Line 246) but doesn't show it. A zoom in mesh is necessary for this.**
>
> We included zoom-in views of textured geometry in Fig. 4 to illustrate local topology changes. We will provide additional zoom-in views of mesh geometry in the appendix.
>
> **Q11: What are the mesh sizes and dimensionalities of printed objects?**
>
> The average number of vertices/elements per generated mesh is 27k/54k. We remark that our mesh resolution, determined by the underlying implicit representation, is adjustable based on user needs. Additionally, we can optimize a low-resolution mesh and output a final mesh with higher resolution.
>
> **Q12: Are Figure 7 results printed?**
>
> We did not print them.
>
> **Q13: x_{com} is not defined in Eq. 7.**
>
> It denotes the position of the center of mass of the underlying (solid with uniform density) geometry; we’ll clarify this in the revised version.

---

> > ### Comment · Reviewer_iQck · 2024-08-10
> >
> > Thank you for your explanations. I changed my score and agree with accepting the paper.

---

### Official Review · Reviewer_XTuH · 2024-07-13

**Soundness:** 3
**Presentation:** 3
**Contribution:** 3
**Rating:** 6
**Confidence:** 4

**Summary:**

This paper introduces a method to make 3D assets produced by SDS-based generative models stand on their own. On top of the SDS loss, it proposes a “standability" loss that encourages the rigid-body simulation result to be rotation-free, and a “stable” loss to encourage the generated shape to be a “local minimum” of height for the center of mass. The evaluation is conducted both in simulation and 3D-printed shapes in the real world.

**Strengths:**

- This paper addresses an interesting problem of making 3D-generated shapes stand. Although this problem has been studied from the perspective of 3D printing, this paper introduces new methods tailored to the new setting of SDS-based 3D generation.
- The proposed method seems to be effective from the various presented results, with a significant improvement in the success rate of standing.

**Weaknesses:**

- The proposed method involves solving an optimization that involves multiple loss terms. No ablation study is provided to prove the necessity of each term. In particular, are both standability loss and stable loss required? How are their influence on the optimization results respectively? Are both normal smoothing Eq (8) and Laplacian smoothing Eq (9) required and why is Laplacian smoothing only applied to the bottom of the shape?
- The traditional balancing method in computational fabrication [59] is only discussed but not actually compared in experiments.
- What are the loss weights for each term in Eq (10)? These numbers may be tricky to set, and thus make the paper hard to reproduce.
- How is \(T\) set in Eq (5)? How do different choices of \(T\) influence the results?
- The success rate of the baseline method is not reported in Fig. 6 for the audience to understand the improvement.

**Questions:**

See weaknesses. These are important questions to answer for a well-rounded paper.

**Limitations:**

Seems adequate.

---

> ### Author Rebuttal · Authors · 2024-08-05
>
> We thank you for recognizing the novelty and effectiveness of our work. Indeed, our work is not directly targeting 3D printing, but an automated 3D generation pipeline without manual tuning.
>
> **Q1: No ablation study is provided to prove the necessity of each term. In particular, are both standability loss and stable loss required? How are their influence on the optimization results respectively? Are both normal smoothing Eq (8) and Laplacian smoothing Eq (9) required and why is Laplacian smoothing only applied to the bottom of the shape?**
>
> We perform additional ablation studies, shown in Fig. 3 in the attached PDF, to justify the necessity of each term. In particular, our standability loss is necessary for ensuring successful self-support, as it uses physical simulation for verification (Fig. 3 (b)); the stability loss is included to ensure that the object remains stable under small perturbations (Table 1), since, in real-world applications, objects may not be placed in the exact same initial pose; our Laplacian smoothing is applied to avoid artifacts such as irregular shapes on the contact surface (Fig. 3 (d)), and is therefore only applied at the bottom; while the normal smoothing loss is used to improve the overall appearance of the shape, as the mesh generated in the coarse stage is usually very rough.
>
> **Q2: The traditional balancing method in computational fabrication [59] is only discussed but not actually compared in experiments.**
>
> We provide additional comparison experiments with make-it-stand [59], as shown in Fig. 2 in the attached PDF. In our global response, we discuss in detail the drawbacks of such post-processing methods compared to our joint optimization approach. Please refer to the global response for more details. Overall, post-processing methods may overlook semantics during optimization, leading to degraded text alignment. In particular, make-it-stand [59] assumes the contact region is unchanged. Such inflexibility may result in undesired outcomes, as demonstrated in Fig. 2. In addition, make-it-stand [59] requires more human intervention, such as selecting multiple thresholds and placing handles, whereas our approach is more streamlined.
>
> **Q3: What are the loss weights for each term in Eq (10)? These numbers may be tricky to set, and thus make the paper hard to reproduce.**
>
> In our experiments, we use the following weights for our loss terms by default: {lam_sds = 1, lam_normal = 1e4, lam_stand = 1e5, lam_stable = 1e5, lam_blap = 1e7}. For a few examples, we tune these weights within the following ranges: {lam_stand = 1e5-5e5, lam_stable = 1e5-5e5, lam_blap = 1e6-1e7}. Our heuristic intuition is to keep the SDS and physical loss terms roughly on the same scale. For the regularization terms, we scale them to around 1/1000 to 1/100 of the SDS and physical loss terms. We will release our code upon acceptance so the community can reproduce our results.
>
> **Q4: How is (T) set in Eq (5)? How do different choices of (T) influence the results?**
>
> We empirically set $T = 2$ seconds, which in practice is sufficient to generate standable results. Setting $T$ to a larger value will not alter the simulated pose at the end time if the model has reached a steady state, and thus will not affect the resulting loss. However, it will significantly increase the simulation and the subsequent loss backward time, reducing computational efficiency. Conversely, setting $T$ to a smaller value may not be long enough to capture the final steady state, resulting in insufficient penalization of instability and ultimately leading to unstandable 3D models.
>
> **Q5: The success rate of the baseline method is not reported in Fig. 6 for the audience to understand the improvement**
>
> When verified in simulation, none of the generated results from the baseline method can stand stably when placed straight up even without perturbation (as mentioned in lines 276-279). In other words, the success rate of baseline methods under perturbation is zero.

---

### Official Review · Reviewer_MwTq · 2024-07-14

**Soundness:** 3
**Presentation:** 3
**Contribution:** 2
**Rating:** 5
**Confidence:** 3

**Summary:**

The paper introduces a differentiable simulation-based loss to refine the existing SDS(Score Distillation Sampling)-based text-to-3D frameworks. Concretely, it relies on a differentiable simulator Warp to provide gradients for keeping the rotation of the generated mesh unchanged after a period of time. Besides, the authors also introduce a stable equilibrium loss, which favors a flat contact surface. Experiments show that the proposed method can generate self-supporting 3D models given text prompts, verified by simulation and real-world results with 3D-printed objects.

**Strengths:**

The paper introduces an effective and flexible method to refine the existing SDS(Score Distillation Sampling)-based text-to-3D frameworks, like Magic3D and MVDream. The authors also provide experiments on real-world validation.

**Weaknesses:**

1. The paper lacks enough baselines to show that the task the paper tackles is non-trivial. For example, the claim that "Directly integrating these methods with 3D generative AI as a postprocessing module is suboptimal" (L55-L56) is not verified by the experiments. If a self-supporting mesh is needed, the simplest baseline can be cutting the mesh by a flat plane moderately higher than the lowest vertex, as a postprocessing step.
2. The physical constraints used in this work are relatively limited. The standability loss only keeps the rotation unchanged. It can not recover some common cases like "a standing horse" (e.g., Napoleon Crossing the Alps), although those cases may be beyond "self-supporting" defined in this paper. It seems that the simulator has not been fully leveraged, since a simple post-processing baseline (e.g., flatting surfaces that contact with the ground) may also achieve good standability and stability.
3. The current method is only applied to SDS-based methods only (already pointed out by the authors in the limitations)

**Questions:**

1. Can the authors provide more baselines, especially any existing post-processing method with the help of differentiable simulators, to enhance their technical contributions? One example is given in the *Weaknesses* section.
2. Can the authors give a clear definition of "self-supporting"? It is a little ambiguous, since one can replace the initial pose with the pose after simulation so that Eq. (3) about standability can be satisfied, as long as the limit exists.

**Limitations:**

The paper has addressed its limitations.

---

> ### Author Rebuttal · Authors · 2024-08-05
>
> We thank you for recognizing the effectiveness and flexibility of our framework. Indeed, our pipeline can work with different 3D generative models and different differentiable physical simulators, allowing potential variants and extensions.
>
> **Q1: The paper lacks enough baselines to show that the task the paper tackles is non-trivial [...], the simplest baseline can be cutting the mesh by a flat plane moderately higher than the lowest vertex [...] Can the authors provide more baselines?**
>
> We show more baseline results in the attached PDF. Cutting the mesh by a flat plane moderately above the lowest vertex cannot robustly generate a standable output.  As shown in Fig. 1 in the attached PDF, for the case of “a standing goose,” horizontally cutting the mesh from the bottom at various heights all result in failure especially when the center of mass is outside the supporting region. We additionally compare with *make-it-stand* [59]. As shown in Fig. 2 in the attached PDF, *make-it-stand* [59] holds the supporting surface unchanged, potentially leading to distorted results. Additionally, its results suffer from worsened text alignment due to the overlook of semantics.
>
> **Q2: The standability loss only keeps the rotation unchanged. [...] It seems that the simulator has not been fully leveraged, since a simple post-processing baseline (e.g., flatting surfaces that contact with the ground) may also achieve good standability and stability.**
>
> In this work, we consider the 3D models as rigid bodies whose dynamical states can simply be represented by rotation $R$ and translation $T$. We only consider the rotation $R$ in the standability loss as real-world instability mostly leads to rotational deviation from the initial state. Motion related to translation, such as falling due to gravity, is irrelevant to standability and is therefore disregarded. This point is explained in Line 159-161 of our paper. For future exploration involving other types of materials, such as soft bodies, additional physical state parameters like deformation may need to be considered.
> As shown in Fig. 1 and 2 in the attached PDF, compared with the post-processing baselines, like cutting the bottom and make-it-stand [59], our joint optimization with simulation loss can dynamically adjust the center of mass and the contact region without compromising the text alignment. Please refer to our global response for more details on joint optimization versus post-processing.
>
> **Q3: Can the authors give a clear definition of "self-supporting"? It is a little ambiguous, since one can replace the initial pose with the pose after simulation so that Eq. (3) about standability can be satisfied, as long as the limit exists.**
>
> As elaborated in our global response, we set the initial pose (upright direction) based on the output of text-to-3D generative models, as the upright direction is semantics-driven and generative models inherently learn these semantics from their training data. For the definition of self-supporting, we consider it as the condition where, once the upright pose is determined by the text-to-3D model, the generated 3D model should satisfy the standability criteria defined in Eq. (3).

---

> > ### Comment · Reviewer_MwTq · 2024-08-11
> >
> > First of all, thank you for the answers in the rebuttal.
> > For Q1, I appreciate that the authors provide comparison on certain, perhaps representative, cases qualitatively. However, it is better to compare with other methods quantitatively, which will make the paper more convincing. I would like to keep my rating.

---

### Official Review · Reviewer_Nrjn · 2024-07-15

**Soundness:** 4
**Presentation:** 4
**Contribution:** 4
**Rating:** 8
**Confidence:** 4

**Summary:**

This paper addresses the problem of generating 3D models from text that are visually appealing but often physically unstable in simulations or when 3D printed. The authors incorporates a differentiable simulation-based loss function and physically inspired regularization to ensure generated models are stable under perturbation. The proposed method involves a two-stage training process where a coarse model is first generated from text prompts and then refined with physical constraints to ensure stability. Experiments show that models created with Atlas3D maintain stability better than those from existing methods. The optimized models are also 3D fabricated to verify the stability in real world.

**Strengths:**

The paper proposes an original and important solution to the issue of physical stability in text-to-3D generation, combining differentiable simulation-based loss functions with physically inspired regularization. This approach significantly enhances existing methods, reducing the need for manual post-processing and making the models immediately practical for various applications.

The method is sound, with clear and concise writing that guides the reader effectively and should be straightforward to implement for someone with a background in physical simulation. The experiment quality is high, validated through extensive experiments in both simulations and real-world scenarios, demonstrating the method's robustness and versatility. The results confirm the method's effectiveness in ensuring the stability of 3D models.

**Weaknesses:**

The paper is strong overall, and I did not identify any significant weaknesses. However, to further support future research in this area, I would suggest releasing the code. Providing the implementation would greatly benefit the community, enabling others to replicate the results and build upon this work.

**Questions:**

I don't have any specific questions at this time. The paper is clear and well-executed, addressing an important issue effectively. Implementation is also described in detail.

**Limitations:**

I am curious about the failure modes of Atlas3D. Could you provide more details on the scenarios or conditions under which the method might not perform as expected? Understanding these limitations would be helpful for future research and applications. Additionally, addressing these potential failure modes could further strengthen the robustness of your approach.

---

> ### Author Rebuttal · Authors · 2024-08-05
>
> We thank you for recognizing the novelty and effectiveness of our method. To the best of our knowledge, our method is the first to bring standability to large generative models via joint optimization of standability loss and score distillation sampling loss.
>
> **Q1: I am curious about the failure modes of Atlas3D. Could you provide more details on the scenarios or conditions under which the method might not perform as expected?**
>
> Our approach may fail under certain cases. Firstly, some text prompts inherently contradict the concept of standability, such as “a swan and its cygnets swimming in a pond” or “a beautiful rainbow fish.” In these cases, the SDS loss may conflict with the physical loss, leading to unsatisfactory results. Additionally, since we assume models are solid with uniform density, under certain geometries where the projected position of the center of mass onto the horizontal plane is way off the contact region, our approach may not converge to the global minimum of the standability loss (which is zero).
>
> **Q2: I would suggest releasing the code.**
>
> Thank you for the suggestion. Our code has been cleaned up and is ready for release upon acceptance.

---

> > ### Comment · Reviewer_Nrjn · 2024-08-13
> >
> > Thank you for the response. I do not have further questions.

---

### Author Rebuttal · Authors · 2024-08-05

Dear Reviewers,

We would like to thank all reviewers for your insightful and constructive feedback. We are encouraged by the recognition that our paper:
- Addresses the interesting problem of physical stability in text-to-3D generation and provides an effective and important solution [Reviewer Nrjn, MwTq, XTuH]
- Introduces elegant standability and stable equilibrium losses [Reviewer iQck]
- Includes well-validated experiments in both simulation and real-world settings [Reviewer Nrjn, MwTq, XTuH, iQck]

Below, we first address some common questions raised by the reviewers.
## Upright Direction
We believe that the choice of the upright direction is semantically driven and requires human judgment. Different people might have different interpretations of what an upright pose should be. Our objective is to enable the automatic generation of self-supporting 3D models without human intervention, including manually selecting the upright direction. Existing large pre-trained 3D generative models are based on human-made assets, typically crafted to be upright by artists. By learning from such data, these models inherently have an understanding of the upright direction. Thus we designate the vertical direction from the default 3D generation output as the upright direction. This also allows batch evaluation of our method, as shown in Fig. 8 in our paper..
## Joint Optimization v.s. Post-processing
While directly post-processing 3D generated models is a straightforward and effective approach to achieving physical stability, it may result in outcomes that do not align with the text prompt, as it overlooks semantics. One simple post-processing method, as pointed out by Reviewer MwTq, is to cut the mesh by a flat plane slightly higher than the lowest vertex. However, this method will fail when the projection of the center of mass lies outside the contact region, as shown in Fig. 1 in the attached PDF. Additionally, determining the cutting height is another parameter that must be manually set or optimized, and this may also degrade the overall appearance.
Another post-processing method, *make-it-stand* [59], offers an effective way to relocate the center of mass to achieve standability. However, it assumes that the supporting surface is fixed during optimization, which can lead to distorted results due to the imperfect quality of text-to-3D generated models. We provide two examples in Fig. 2 in the attached PDF to illustrate this point. For the goose example generated by Magic3D, one leg is shorter than the other. *Make-it-stand* only treats one foot as the supporting surface and ignores the other due to its post-processing nature, whereas our joint-optimization pipeline enables stable standing with two legs on the ground. A similar issue also happens to the kangaroo example. More importantly, the text alignment degrades as semantics are overlooked during optimization. In contrast, our proposed joint optimization method preserves the text alignment and dynamically adjusts the center of mass as well as the supporting surface configuration. In Fig. 3(e) in the attached PDF, we also apply our proposed losses in a post-processing manner. Similarly, while it is able to optimize the 3D models to make it standable, the text alignment is compromised. Overall, compared to the post-processing method, joint optimization is a more robust way which better balances text alignment and physical constraints.
## Ablation Study
We perform additional ablation studies in the attached PDF to demonstrate the necessity of our proposed losses. It can be observed in Fig. 3(b), that without standability loss $L_\text{stand}$, the figure fails to stand. While the figure can still stand without stable equilibrium loss $L_\text{stable}$, as demonstrated in Fig. 3(c), it is less stable under perturbation. As shown in Tab. 1, introducing the stability loss consistently increases the success rate of standing under different scales of perturbations. Additionally, we show the effectiveness of geometry regularization loss term $L_\text{b-lap}$ in Fig. 3(d), which helps smooth the geometry and avoid spiky artifacts on the surface. We will incorporate these results into our revised version.

We believe that our pipeline is lightweight and is applicable to a wide range of 3D generative models. We will release our code upon acceptance, as suggested by Reviewer Nrjn.

We also provided detailed responses to each reviewer separately.

---

### Decision · Program_Chairs · 2024-09-25

**Decision:**

Accept (poster)

**Comment:**

This paper received divergent initial opinions with three reviewers in support of acceptance and one negative reviewer.  The main concerns by the negative reviewer revolved around missing ablations to demonstrate the impact of loss terms, and a lack of clarity about relation to robotics and fabrication.  The rebuttal responded to reviewer concerns, and the initially negative reviewer raised their opinion to accept.
 The AC does not find a basis to overrule the reviewer consensus in favor of acceptance.  The AC thus recommends acceptance and strongly encourages the authors to incorporate clarifications and revisions to improve the final manuscript.